# Experimental Study on Whole Wind Power Structure with Innovative Open-Ended Pile Foundation under Long-Term Horizontal Loading

**DOI:** 10.3390/s20185348

**Published:** 2020-09-18

**Authors:** Junwei Liu, Zhipeng Wan, Xingke Dai, Dongsheng Jeng, Yanping Zhao

**Affiliations:** 1College of Architecture and Civil Engineering, Qingdao University of Technology, Qingdao 266033, China; liujunwei@qut.edu.cn (J.L.); zhipengwan0@gmail.com (Z.W.); daixingke@gmail.com (X.D.); 2School of Engineering and Built Environment, Griffith University, Gold Coast Campus, Southport QLD 4222, Australia; 3Shandong Bureau of China Metallurgical Geology Bureau, Jinan 250014, China; zhaoyanping06@gmail.com

**Keywords:** OWE pile foundation, horizontal cyclic loading, horizontal displacement, maximum moment

## Abstract

The offshore wind energy (OWE) pile foundation is mainly a large diameter open-ended single pile in shallow water, which has to bear long-term horizontal cyclic loads such as wind and waves during OWE project lifetime. Under the complex cyclic loads, the stress and displacement fields of the pile-soil system change continuously, which affects the dynamic characteristics of the pile foundation. Within the service life of the pile foundation, the pile-soil system has irreversible cumulative deformation, which further causes damage to the whole structure. Therefore, it is important to examine the overall dynamic characteristics of wind power foundation under high cycle. In this paper, in the dry sand foundation, taking the Burbo Bank 3.6 MW offshore turbine-foundation structure as the prototype, the horizontal cyclic loading model tests of the wind power pile foundation with the scale of 1:50 were carried out. Considering the factors such as loading frequency and cyclic load ratio, the horizontal dynamic characteristics of the whole OWE pile foundation are studied. The comparison results between the maximum bending moment of pile and the fitting formula are discussed. In conclusion, moment of OWE pile shaft is corresponding to the loading frequency (*f* = 9 HZ) and loading cycles by fitting formulas. The fatigue damage of the OWE pile does not occurs with low frequencies in high cycles.

## 1. Introduction

To date, large diameter open-ended single pile foundations in shallow water [1] have been used as the supporting structures for 75% of OWE projects [2]. Because of its mature construction technology, low cost, and better benefit, single pile foundation has occupied a large market and broad application prospects in offshore wind power generation [3,4]. However, the whole OWE structure is affected by wind, wave, current, and other complex loads [5]. These complex loads change the internal forces and displacements of the OWE pile foundation, which brings difficulties in the design and construction of the pile foundation. Hence, in numerous design codes [6,7], it is required to evaluate the pile displacement and the maximum bending moment, and reduce the total deformation in the design. Therefore, it is necessary to consider the influence of complex load amplitude, frequency, and other parameters on the deformation and internal force of the pile foundation [8].

In the early stage, Ref. [9] introduced the concept of pile deformation and pile internal force in the loading stage, and carried out a series of experimental studies. Later, Refs. [10,11] conducted a series of field tests and model tests of pile foundation in clay and sand foundation under horizontal cyclic loading. Based on their experimental results, they proposed the *p-y* curve method, which has been widely accepted by practical engineers. In the past decade, numerous experimental studies for the horizontal loading characteristics of piles in sand have been carried out [12,13,14]. In addition to 1g tests, Refs. [15,16,17] carried out a centrifuge model test on the horizontal loading characteristics of piles in sand. Through their tests, the cumulative deformation law of displacement and internal force of pile under cyclic load was established, and the expression of cumulative deformation of displacement and internal force of pile foundation under horizontal loading was obtained.

However, most previous model tests were often a small-scale test with a small number of cycles and a single loading order. Furthermore, small-scale tests have significant influence on the boundary effect, which cannot always be ignored, while the centrifuge test cannot be loaded synchronously. Meanwhile, there are problems in the observation and quantification of experimental phenomena. For example, the existing research cannot describe the internal force and displacements of the pile with the combination of load frequencies, load amplitudes, and vibration times well. Thus, it is difficult to well deduce the close relationship between displacement and internal force in the loading process of the pile.

Recently, Refs. [18,19,20,21] equaled the offshore single pile, wind tower, and other structures to a pile foundation with mass at the top of the pile, simplifying the analysis of the OWE pile foundation. Ref. [22] found that the inclusion of the interaction of soil-structure has a significant influence on the dynamic characteristics of OWE foundation system, and the first-order natural frequency of soil-structure is reduced. The description of complex pile-soil interaction plays a key role in the internal force and deformation of pile, such as the proper definition of external forcing and the non-linear interactions [23,24]. Later, based on FEM model (PLAXIS3D 2013, PLAXIS B.V.: Exton, PA, USA), Ref. [25] further designed and analyzed the wind power foundation, and examined the importance of the soil-structure interaction. As reported in [26], some unfavorable combinations of structural parameters and soil parameters in the design OWE foundation will affect the dynamic characteristics of the whole system. Therefore, it is necessary to conduct a comprehensive dynamic analysis of the stability of the wind power foundation.

The aim of this study is to further explore the influence of horizontal loading on the internal force and displacement of pile, especially when loading frequency is equal to the natural frequency. In this paper, on the basis of predecessors, a large-scale model test is selected to discuss the law of internal force and deformation of pile under load from a macroscopic point of view. The experimental set-up will be outlined first, including simplified OWE pile foundation and instrumented model box.

## 2. Experimental Set-Up

In the complex interaction process between pile and soil, there are problems such as bearing capacity degradation, service failure [27,28]. However, 1 g model test is still a reliable and common test methodology. Therefore, this study carries out the relevant research on the dynamic characteristics of the whole structure of wind power pile foundation under long-term lateral loading in the 1 g large-scale model test. In this section, the details of experiments are provided.

### 2.1. Soil Sample Preparation

As the dynamic interaction between the pile foundation and the soil around the pile is complex, to reduce the influence of secondary factors on the test, dry sand is used as the foundation soil. The soil samples were naturally air-dried Qingdao sea sand which was directly obtained from the field. The layered compaction method is adopted, and the samples are compacted with the relative density of 0.7. Compaction is carried out once per 10 cm until the height of the foundation meets the test requirements. The physical parameters of sand are obtained by the standard soil mechanics tests in the laboratory. In the instrumented model box, the laser displacement transfers (LDT) were used to monitor the displacement of the pile, and strain gauges were arranged inside and outside the pile to measure the variation of internal force of the pile. The particle gradation curve of sand sample is shown in Figure 1, and the physical parameters of Qingdao sand are given in Table 1. The model test in this paper used dry sand and did not involve the similarity of drainage [20,29].

### 2.2. OWE Foundation Model

According to the similarity theory, this study used the Burbo Bank 3.6MW (4C Offshore Limited., GE Energy, UK, 2017) offshore structure as the prototype. The scale of 1:50 is adopted for the design of experiments. The penetrated depth of the soil is 600 mm. The model consists of four parts: turbine which is simplified to a large mass tower, pile, model chamber, and uniform sand. To make the model more consistent with the actual structure of OWE foundation, the tower is connected by two sections of aluminum tubes. One section is 1000 mm and the other is 700 mm, that is, the total height of the tower is 1700 mm. The vertical distance from the top of tower to the mud surface is 2100 mm. In this study, the concentrated mass is applied to the top of the tower by counterweight. The loading point is at the 1800 mm above the mud surface. For the selection of model pile materials, the principle of physical condition similarity should be satisfied: C_σ_ = C_E_C_ε_, where C_σ_ is stress similarity ratio; C_E_ is elastic modulus similarity ratio; C_ε_ is strain similarity ratio. The Poisson’s ratio of 6063 aluminum alloy pipe is close to that of steel pipe. The main parameters of the model are listed in Table 2 and Table 3. Figure 2 and Figure 3 show the schematic of the whole structure of wind power pile foundation and model test, respectively.

### 2.3. Design of Loading

Based on the principle of centrifugal force, a horizontal loading device is made, as shown in Figure 4. The loading device can adjust the loading force by changing the mass and relative position of M1 (mass 1) and M2 (mass 2). The loading frequency can be measured by a tachometer.

The loading device is bidirectional loading, and its loading mode is defined as shown in Figure 5. The degree of cyclic load applied is expressed by defining *ζ*_b_ = *H*_max_/*H*_u_ [13]. *H*_u_ represents the horizontal ultimate static bearing capacity of the whole structure of wind power pile foundation and *H*_max_ represents the maximum load in a cycle. *f* is the frequency.

The purpose of this test is to examine the dynamic response of the whole structure of wind power pile foundation under long-term cyclic loading. Because the number of cyclic loading is more than 100,000 times, therefore, it is particularly important to design a feasible frequency range for the dynamic response of the model. The loading frequency used in the model test is “*1P*” (exciting force frequency caused by the blade rotation) or “*3P*” (exciting force frequency caused by the “shielding effect” of blade rotation on frame tower) [30]. 

The “*1P*” frequency range in the prototype of the whole structure of wind power pile foundation is 0.082–0.216 Hz, the “*3P*” frequency is 0.251–0.652 Hz, and the natural frequency of the wind power foundation prototype is about 0.289 Hz [3]. It is measured that the natural frequency of the whole structure of wind power pile foundation is about 9 Hz.

The frequency of the load applied in the model test is obtained by the equation as follows:(1)$${\left(\frac{f}{f_{n}}\right)}_{model}={\left(\frac{f}{f_{n}}\right)}_{prototype}$$

According to the similarity relationship, the range of “*1P*” is 2.55–6.68 Hz and the range of “*3P*” is 7.7–20 Hz. Based on this, in the model test, the loading frequency is selected to be 6 Hz (*1P*), 9 Hz (natural frequency), and 12 Hz (*3P*), respectively. *f_n_* is the on-site load frequency and *f* is the natural frequency.

### 2.4. Specific Test Scheme

#### 2.4.1. Horizontal Static Load Test

Figure 6 shows the load-displacement curve of horizontal static tests. The displacements of the two groups of tests increase continuously in the process of load application. Those mainly come from two parts, one is the displacement caused by the compression deformation of the soil at the pile-soil interface, and the other is the bending deformation of the pile. There is no obvious inflection point or turning point, which shows that the curve of the whole structure under horizontal static load is generally processing hardening type. That is, large diameter single pile can continue to bear the load when the displacement changes greatly, which will cause huge deformation of the foundation. In this test, the load is taken as the ultimate bearing capacity, when the displacement of the loading point in the load-displacement curve achieves 0.1D (pile diameter) [31]. According to the horizontal static load test, the horizontal ultimate bearing capacity of the whole structure of OWE pile foundation is 189.7 N.

#### 2.4.2. Test Scheme

The main focus of this study is the bearing characteristics of OWE foundation under non-extreme conditions, the test conditions are tabulated in Table 4. The cycle ratios of the tests are 0.1, 0.2, and 0.3, respectively. 6 HZ (*1P*), 9 HZ (the natural frequency), and 12 HZ (*3P*) were selected as the test frequencies respectively. The number of the loading cycle is 1 × 10^5^. The whole scheme is shown in the Table 4.

## 3. Discussion of Test Results and Analysis

### 3.1. Variation of Displacement

As shown in Figure 7 and Figure 8, the changes of displacements and fitting curve parameters at the tower top and the mudline under different loading conditions are given respectively. 

Based on the comparative analysis of the results, the following conclusions can be drawn. 

The displacement value of: “*3P*” is larger than that of “*1P*”and natural frequency. This reflects that “*3P*” has a greater influence on the whole structure of wind power pile foundation. To a certain extent, the cumulative displacement of wind power foundation increases as the load amplitude and frequency increases. When the load ratio is equal to 0.1 and 0.3, the displacement values of the three test natural frequencies are quite different, and the load is the dominant factor of the displacement; when the load ratio is equal to 0.2, the difference of the displacement values of the three test frequencies is small, and the frequency value has little effect on the displacement.

However, it can be inferred from the data that the variation value of the displacement at the top of the tower in sand is larger than that on the mud surface. According to the loading failure standard (the horizontal displacement of pile mud surface is more than 0.1D) of the test, the whole structure of wind power pile foundation is not prone to fatigue failure in the case of small amplitude load ratio. That is,
(2)$$\frac{\Delta_{H}}{L_{P}}<\frac{\Delta_{P}}{L_{F}}<0.1D$$
where Δ*_P_* is cumulative maximum displacement for the top of the tower, *L_F_* is the overall length of tower frame-pile, Δ*_H_* is cumulative maximum displacement of pile at mud surface, and *L_P_* is pile length.

To date, a better method to predict the cumulative lateral displacement of large diameter single pile is to establish the relationship between the extreme displacement and the number of cycles of the cyclic loading. In this study, a power exponential relationship between the cumulative lateral displacement and the number of cycles is adopted. It is considered that the horizontal cumulative displacement of the pile under the number of *N*th cycles is related to that of the number of 100th cycles (a previous specific cycle is selected). The relationship is as follows:(3)$$y_{N}=y_{B}(1+A\cdot (f\cdot \zeta_{b})\cdot \ln(\frac{N}{B}))$$
where *y_B_* is taken as the horizontal cumulative displacement change after 100 cycles, *y_N_* is the horizontal cumulative displacement change after *N* cycles; *A* is the weakening coefficient under cyclic loading; B∈N+, here *B* = 100; *f* is taken as the frequency value; and *ζ_b_* is the cyclic load ratio.

The cumulative displacement at the top of the whole structure of OWE pile foundation and the cumulative displacement of the mud surface are fitted, and the range of *A* (cycle weakening coefficient) is 0.18–0.24. The expression of *A* is as follows:(4)$$A=\{\begin{array}{c} 0.24,\zeta_{b}=0.1\\ 0.18,\zeta_{b}=0.2\;or\;0.3 \end{array}$$

### 3.2. Variation of Bending Moment of Pile

In the model tests, the strain gauges are installed on both sides (the connection positions of the two strain gauges are perpendicular to the loading direction), and the strain gauges on both sides satisfy the pure bending at the same horizontal height. According to the relevant theories in material mechanics, the bending moment at any point can be determined by
(5)$$I=\frac{\pi D^{4}}{64}(1-a^{4})$$
(6)$$M=\frac{EI\Delta \varepsilon }{r}$$
(7)$$\Delta \varepsilon =\varepsilon_{1}-\varepsilon_{2}$$
where *I* is the moment of inertia of the pile section; a is inner diameter divided by outer diameter; *r* is the distance between the strain measuring point and the neutral layer, that is, the radius; *EI* is the bending stiffness of the pile. *M* is the bending moment of the pile, *ε* is the strain of the pile, and *ε*_1_ (the tension side parallel to the loading direction) and *ε*_2_ (The compressed side parallel to the loading direction) are the strains on both sides of the pile at the same height.

The change of bending moment of pile body has general characteristics in different frequencies. Aiming at studying the variation of moment of pile shaft under natural frequency, 9 HZ is selected to focus on the study. Based on the results of dynamic load test under 9 HZ, the moment variation curve of the test is given as follows.

Figure 9 shows that the position of the maximum bending moment in the pile foundation is basically about 2D (pile diameter) below the mudline. To a certain extent, the bending moment of the pile increases with the increase of buried depth. It will decrease after reaching the maximum. 

With the progress of cyclic loading, the position of the maximum bending moment basically does not change, and the maximum bending moment increases continuously. The maximum bending moment occurs at the distance of about 40 cm from the pile end. The bending moment below the 18 cm from the end of the pile is almost zero.

The load amplitude has a great influence on the maximum bending moment, and the maximum bending moment increases with the increase of the loading amplitude in different frequencies. Generally speaking, the bending moment of the pile varies with the load ratio, and the final bending moment and load ratio increase exponentially. To a certain extent, the change of the bending moment is consistent with the change of the displacement of the mud surface. The bending moment of the tower top shows that measured by *3P* is larger than that measured by *1P*. At the same time, the change of load ratio makes the change of bending moment more discrete, thus, the smaller the load ratio is, the faster the approximate value of the final bending moment is; on the contrary, the slower it will be.

### 3.3. Comparative Analysis of Measured Value and Fitting Value of Maximum Bending Moment of Pile

The variation of the bending moment of the pile foundation has a general trend. Herein, the analysis of the maximum bending moment of the pile foundation with *f* = 9 HZ is taken as an example. It is assumed that the maximum bending moment of the pile (its position in the pile does not change) satisfies the following power exponential relationship:(8)$$M_{T}=\{\begin{array}{c} M_{0}(1+\alpha \cdot {\left(A\right)}^{2}(f\cdot \zeta_{b})\cdot \ln(\frac{T}{B}))\\ M_{0}(1+\beta \cdot {\left(A\right)}^{2}(f\cdot \zeta_{b})\cdot \ln(\frac{T}{B})) \end{array}$$
where *M_T_* and *M*_0_ are the maximum bending moment of the pile under the *N*th and 100th cycles, respectively, and the *A* value is the weakening coefficient, as mentioned earlier. T is the number of cycles. *α* and *β* are correction coefficient of pile bending moment. Here, *α* = 0.7, *β* = 0.3, *α* + *β* = 1.

Figure 10 shows that the fitting envelope can well “wrap” the measured data, which proves the correctness of the fitting formula in this experiment. When fitting the lateral cyclic cumulative displacement and the maximum bending moment of the pile, the cyclic weakening coefficient A is universal, which makes a further relationship between the cyclic cumulative displacement and the bending moment of the pile foundation. In addition, it can be concluded in the diagram that increasing the cyclic load ratio on the basis of (a), (b), and (c) will increase the upper limit of the envelope and decrease the lower limit of the envelope, so that the measured data are always in the envelope.

### 3.4. Degradation Analysis of Bearing Capacity

Figure 11 shows the static calculation curves of each pile after different cyclic loading. It can be seen from Figure 11 that the bearing capacity of the OWE pile foundation attenuates to a certain extent. When the cyclic load ratio is in the range of 0.1–0.3, the attenuation range of open pile foundation at different frequencies is quite different. For example, the horizontal bearing capacity decays more than 10% (under *f* = 6 HZ), 15% (under *f* = 9 HZ), 25% (under *f* = 12 HZ), respectively.

Based on the above comparative analysis, it can be seen that the bearing capacity of the whole structure of OWE pile foundation will decrease after long-term cyclic load. Furthermore, the extent of bearing capacity reduction is different under different cyclic loading conditions. Moreover, continuous loading, higher load ratio, and high frequency loading make the pile have a greater disturbance to the soil around the pile foundation. Consequently, the relative density of the soil is much lower than the initial relative density in long-term loading. As a result, the bearing capacity of the pile foundation decreases with the continuous cyclic loading.

## 4. Conclusions

In this study, a series of experiments for the OWE pile foundation have been conducted. Based on the experimental results, the following conclusions can be drawn as shown below.

(1)Throughout the test results, in the case of small load ratio (*ζ_b_* ≤ 0.3), the cumulative value of the pile foundation is difficult to exceed 0.1D, but the bearing capacity is obviously degraded by more than 10%. With the increase of frequency, the attenuation of the bearing capacity increases.(2)There is a strong correlation between the prediction of the proposed formula for the cumulative displacement deformation and the bending moment.(3)“*3P*” has the largest cumulative effect on pile displacement and deformation, compared with “*1P*.” When the cyclic load ratio is 0.2, the influence of natural frequency on pile displacement and accumulation is approximate to that of “*3P*.”(4)In the process of cyclic loading of the whole structure, the displacement at the top of the tower is larger than the cumulative value at the mud surface. This indicates that the displacement at the top of the tower can be used as an index for the whole structure of OWE pile foundation.

From a macroscopic point of view, this paper studies the changes of pile displacement and bending moment under cyclic loading. The methodology and analysis results are well explained. However, it can be further studied from the following aspects, such as setting different soil conditions (clay, silt, etc.), the formation of pile boots, and the type of cyclic loading force, etc., so that the large-scale model test of OWE pile can be further optimized.

## Figures and Tables

**Figure 1 sensors-20-05348-f001:**
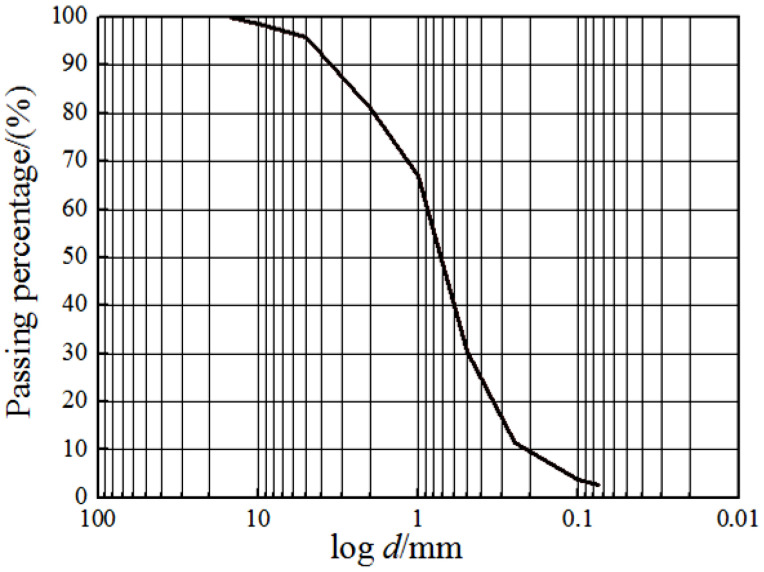
Particle size distribution curve of sand.

**Figure 2 sensors-20-05348-f002:**
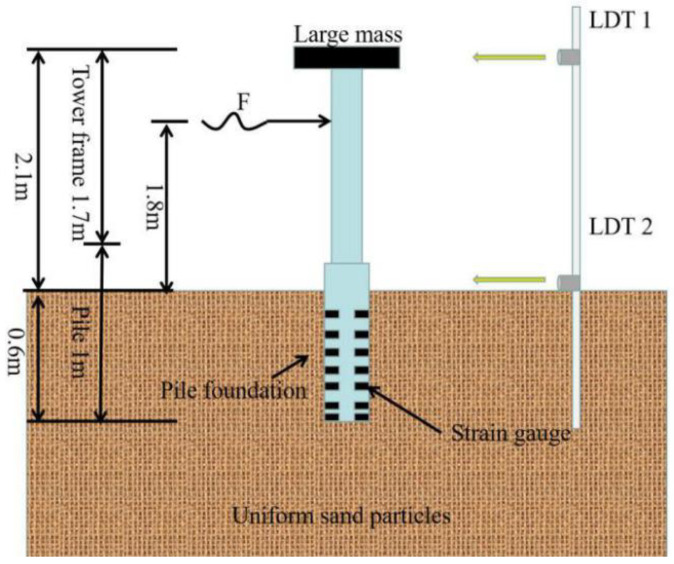
The actual whole structure of wind power foundation. Notes: LDT, a laser displacement transducer; Laser 1 or 2, Laser displacement sensor.

**Figure 3 sensors-20-05348-f003:**
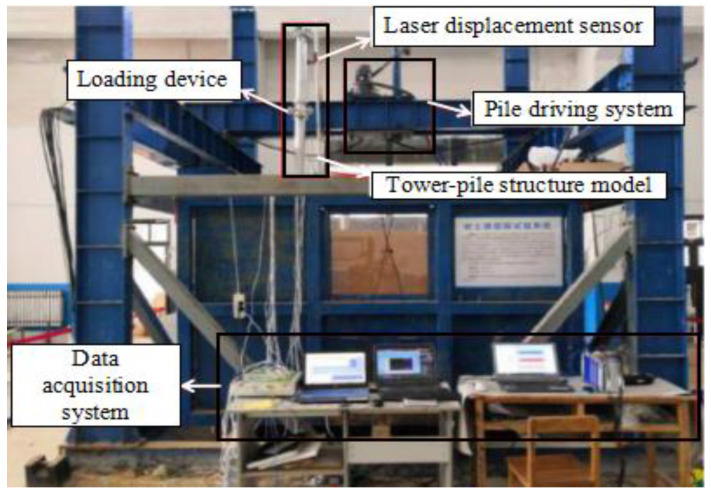
Model test.

**Figure 4 sensors-20-05348-f004:**
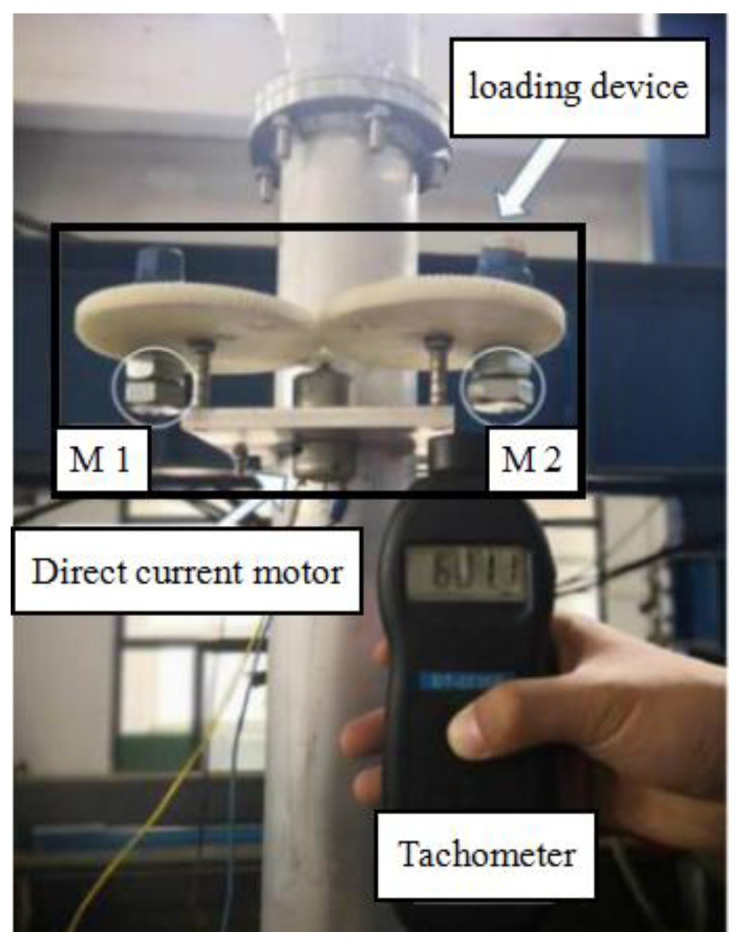
Cyclic loading device.

**Figure 5 sensors-20-05348-f005:**
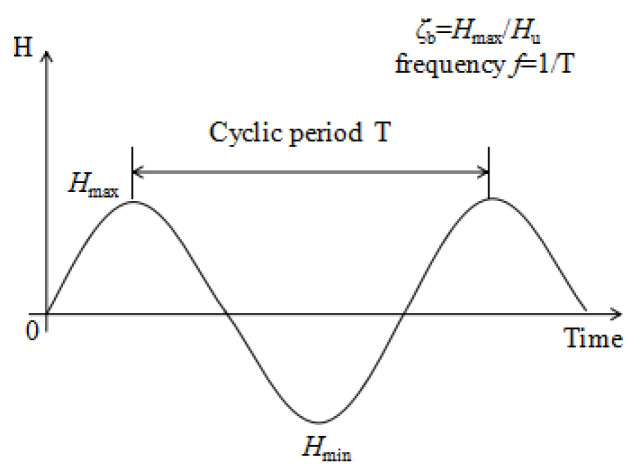
Loading pattern.

**Figure 6 sensors-20-05348-f006:**
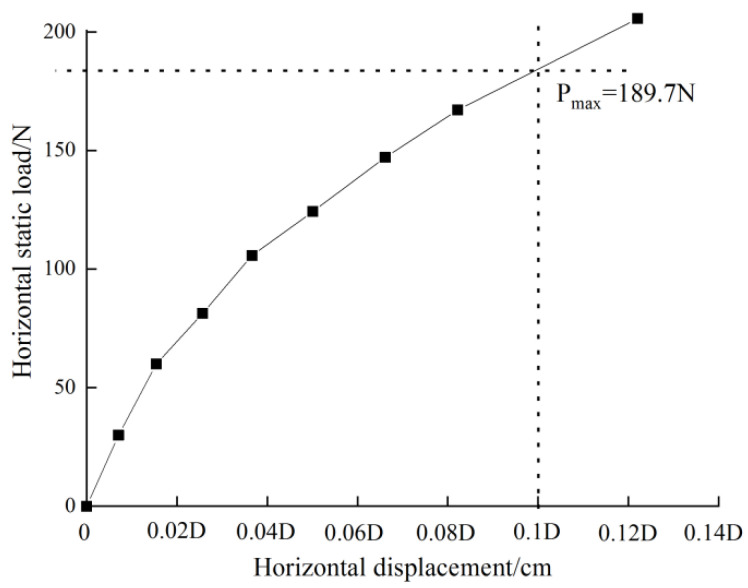
Horizontal static load curve.

**Figure 7 sensors-20-05348-f007:**
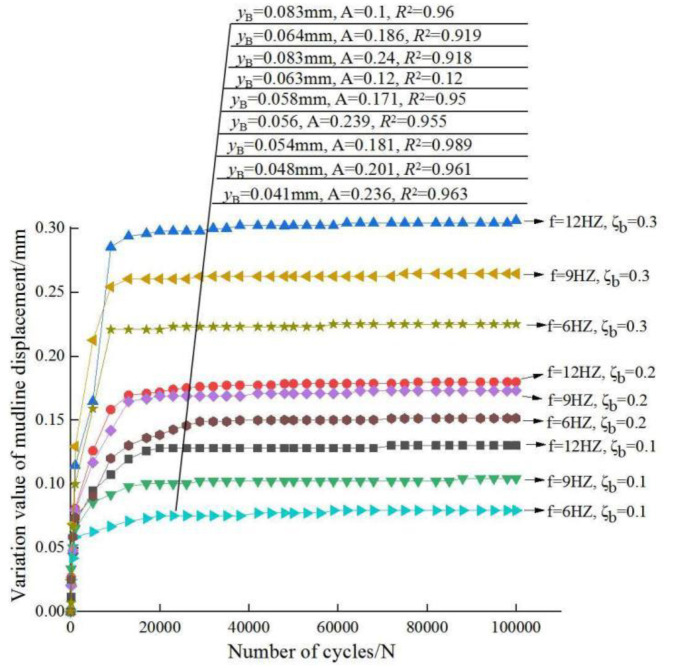
Variation of displacement at the tower top. Notes: The fitting data are consistent with the measured curves from top to bottom and no fitting curve is drawn.

**Figure 8 sensors-20-05348-f008:**
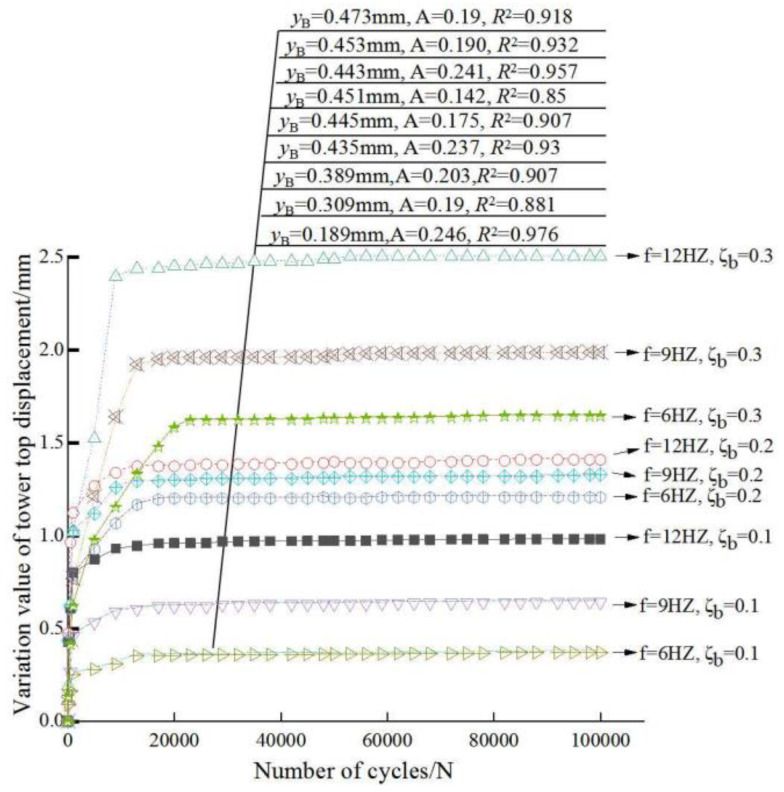
Variation of displacement at the mudline. Notes: The fitting data are consistent with the measured curves from top to bottom and no fitting curve is drawn.

**Figure 9 sensors-20-05348-f009:**
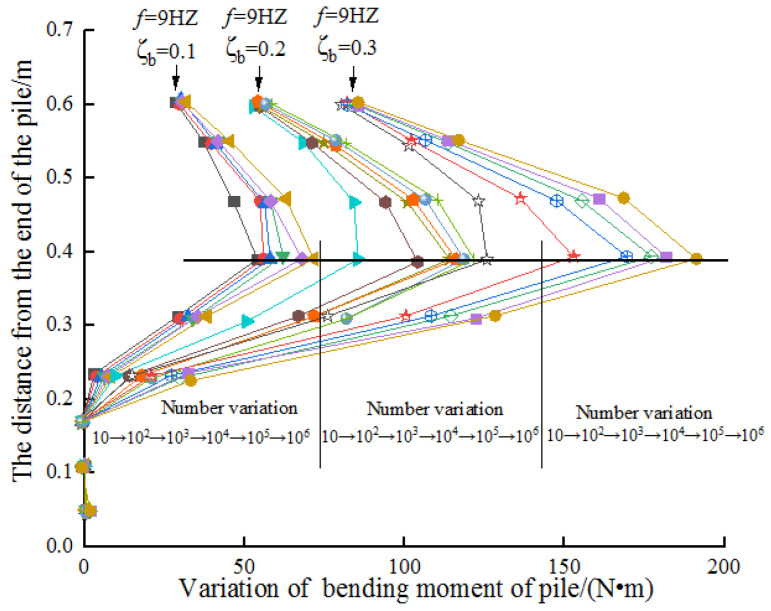
Variation of bending moment of pile.

**Figure 10 sensors-20-05348-f010:**
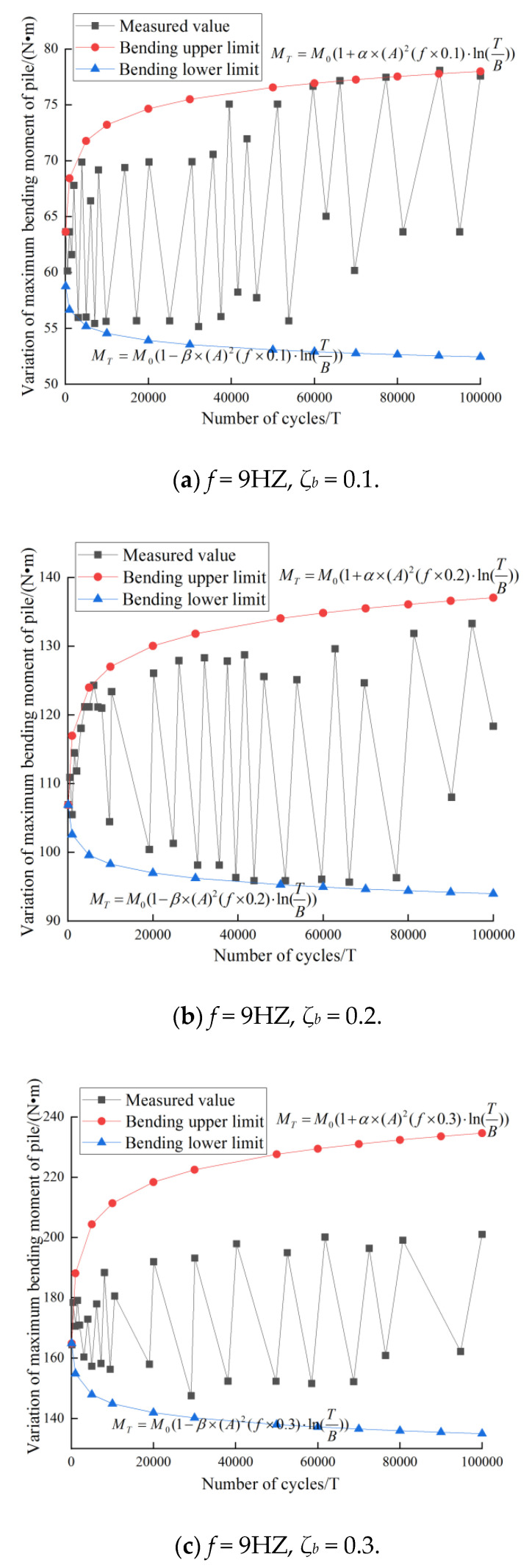
Variation and envelope at the maximum bending moment of pile.

**Figure 11 sensors-20-05348-f011:**
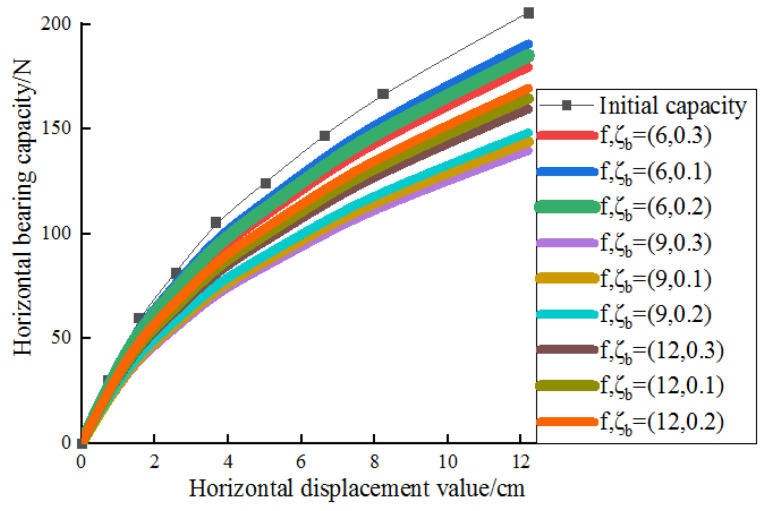
Degradation of static horizontal capacity.

**Table 1 sensors-20-05348-t001:** Physical properties of Qingdao sea sand.

Relative density (*G*_s_)	2.65
Maximum void ratio (*e*_max_)	0.52
Minimum void ratio (*e*_min_)	0.30
Relative compaction (*D*_r_)	0.73
Median particle size (*d*_50_)/(mm)	0.72
Particle size range/(mm)	3~15
Internal friction angle/(Ŷ)	42.8
Dry density (*ρ*_d_)/(*K*g/mm^3^)	1.95

**Table 2 sensors-20-05348-t002:** Properties of model open-ended pile.

Properties	Model Pile
Material	Aluminum
Diameter (D)/(mm)	100
Length (L)/(mm)	1000
Elasticity modulus (E)/(kN/m^2^)	6.9 × 10^7^
Poisson ratio (ν)	0.3
Thickness (t)/(mm)	1.5

**Table 3 sensors-20-05348-t003:** Properties of model tower frame.

Properties	Tower Frame
Material	Aluminum
Diameter (D)/(mm)	93
Length (L)/(mm)	1700
Elasticity modulus (E)/(kN/m^2^)	6.9 × 10^7^
Poisson ratio (ν)	0.3
Thickness (t)/(mm)	1.5

**Table 4 sensors-20-05348-t004:** Scheme of model test.

Test Number	Cyclic Load Ratio	Frequency/Hz	Number of Cycles
Test-1	0.1	6	10^5^
Test-2	0.2	6	10^5^
Test-3	0.3	6	10^5^
Test-4	0.1	9	10^5^
Test-5	0.2	9	10^5^
Test-6	0.3	9	10^5^
Test-7	0.1	12	10^5^
Test-8	0.2	12	10^5^
Test-9	0.3	12	10^5^

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
