# Peer review of "Experimental Study on Whole Wind Power Structure with Innovative Open-Ended Pile Foundation under Long-Term Horizontal Loading"

_sensors, 2020, doi:10.3390/s20185348_

Round 1
Reviewer 1 Report
The paper is about exploring the influence of horizontal loading on the internal force and displacement of OWE pile. The methodology and analysis results are well explained. However, the write up of the paper needs extensive improvements in order to be suitable to publish especially the language in a grammatical sense, phrases selection and sentences structure. I am presenting my remarks and comments in details below:
- There are several grammatical errors and awkward phrases throughout the paper. I recommend full proofreading the paper and use the examples presented below to modify the paper write up.
- There is annoying jump between past and present tense through the paper. I suggest revising and make sure that all the work that was (done) to be in past tense. And all the things the author discovered/verified by their work to be in present tense.
- Line 30 “In a conclusion,” let alone the fact that abstracts does not explicitly mention the paper conclusion. No one writes (in “a” conclusion). Either modify it to “In conclusion” or remove it all together and modify the sentence.
- Line 36 “Instruction” should be changed to “Introduction”.
- Line 53 “which has been a widely accepted by practical engineers.” The current sentence is flawed as one can wonder “a widely accepted (what?) by practical engineers” remove the “a” or modify the sentence to be meaningful in its current form.
- Line 65 “the existing research can not well describe the internal force and displacements” Again the sentence is flawed English. Please change to “cannot describe well” or rephrase.
- Line 87 “reliable and general” did the authors mean “reliable in general”. Please explain.
- Figure 2, the pile and tower should look that the diameters are not the same, the reviewer knows it is a schematic drawing but this makes it easier for the reader to digest the difference between the two parts on the structure.
- Line 122, did the authors mean (laser displacement sensors) but mistakenly wrote (linear displacement sensors). Please clarify. Also, I suggest this notes and all the coming ones to be merged with the paragraph citing the figure.
- Line 164, using bold font which is not consistent with the rest of subsection titles. Please change.
- Table 4 is divided between 2 pages, typically tables should be fit in one page unless the table is larger than one page. Which is not the case here.
- Section 3.1 is written like a report with no logical flow and does not read like the rest of the paper. I highly recommend rewriting the whole section in a more logical and concrete language.
- Lines 226 to 228. Can the authors briefly discuss the moment results from 6 Hz and 12 Hz or state why 9 Hz results only were presented?
- References 20, 22, 24 and 26 are simply the DOI of the previous references. Please correct and re-number the references accordingly.
Author Response
Dear Reviewer#1:
Thank you for the reviewer#1's letter and for your comments concerning our manuscript entitled “Experimental study on whole wind power structure with innovative open-ended pile foundation under long-term horizontal loading” (ID: sensors-905595). Those comments are all valuable and very helpful for revising and improving our paper, as well as the important guiding significance to our researches. We have studied comments carefully have made correction which we hope meet with approval. Revised portion are marked in red in the paper. The main corrections in the paper and the responds to your comments are as flowing:
Responds to the your comments:
1 Line 30 “In a conclusion,” let alone the fact that abstracts does not explicitly mention the paper conclusion. No one writes (in “a” conclusion). Either modify it to “In conclusion” or remove it all together and modify the sentence.
Response:This part of the paper has been revised, showing in the Line 30.
2 Line 35 “Instruction” should be changed to “Introduction”.
Response:This part of the paper has been revised, showing in the Line 35.
3 Line 52 “which has been a widely accepted by practical engineers.” The current sentence is flawed as one can wonder “a widely accepted (what?) by practical engineers” remove the “a” or modify the sentence to be meaningful in its current form.
Response:This part of the paper has been revised, showing in the Line 52.
4 Line 65 “the existing research can not well describe the internal force and displacements” Again the sentence is flawed English. Please change to “cannot describe well” or rephrase.
Response:This part of the paper has been revised, showing in the Line 64-66.
5 Line 87 “reliable and general” did the authors mean “reliable in general”. Please explain.
Response:I mean (common or popular) but mistakenly wrote (general).This part of the paper has been revised, showing in the Line 90.
6 Figure 2, the pile and tower should look that the diameters are not the same, the reviewer knows it is a schematic drawing but this makes it easier for the reader to digest the difference between the two parts on the structure.
Response:This part of the paper has been revised, showing in the Line 122-123.
7 Line 122, did the authors mean (laser displacement sensors) but mistakenly wrote (linear displacement sensors). Please clarify. Also, I suggest this notes and all the coming ones to be merged with the paragraph citing the figure.
Response:This part of the paper has been revised, showing in the Line 125.
8 Line 164, using bold font which is not consistent with the rest of subsection titles. Please change.
Response:This part of the paper has been revised, showing in the Line 156.
9 Table 4 is divided between 2 pages, typically tables should be fit in one page unless the table is larger than one page. Which is not the case here.
Response:This part of the paper has been revised, showing in the Line 184.
10 Section 3.1 is written like a report with no logical flow and does not read like the rest of the paper. I highly recommend rewriting the whole section in a more logical and concrete language.
Response:This part of the paper has been revised, showing in the Line 178-217.
11 Lines 226 to 228. Can the authors briefly discuss the moment results from 6 Hz and 12 Hz or state why 9 Hz results only were presented?
Response:
For the OWE pile, low frequency (1P) is uneconomical for power generation, while the 3P of high frequency will cause damage to the the OWE pile, so the OWE pile is equipped with a set of active transmission system to avoid high frequency.
At this time, the most suitable power generation frequency will appear between 1P and 3P. In the situation, the natural frequency will also fall into it, and the OWE pile is likely to resonate in operation.Therefore, the physical characteristics of pile foundation under natural frequency are emphatically analyzed and discussed.Some references are as follows:
Reference:
- Arany, L. , Bhattacharya, S. , Adhikari, S. , Hogan, S. J. , & Macdonald, J. H. G. . (2015). An analytical model to predict the natural frequency of offshore wind turbines on three-spring flexible foundations using two different beam models. Soil Dynamics and Earthquake Engineering, 74, 40-45.
- Arany, L. , Bhattacharya, S. , Macdonald, J. , & Hogan, S. J. . (2015). Simplified critical mudline bending moment spectra of offshore wind turbine support structures. Wind Energy, 18(12).
- Laszlo Arany, S Bhattacharya, John H G Macdonald, & S John Hogan. (2016). Closed form solution of eigen frequency of monopile supported offshore wind turbines in deeper waters incorporating stiffness of substructure and ssi. Soil Dynamics & Earthquake Engineering, 83, 18-32.
- Arany, L. , Bhattacharya, S. , Macdonald, J. , & Hogan, S. J. . (2017). Design of monopiles for offshore wind turbines in 10 steps. Soil Dynamics & Earthquake Engineering, 92, 126-152.
12 References 20, 22, 24 and 26 are simply the DOI of the previous references. Please correct and re-number the references accordingly.
Response:Here is the problem of format, and these DOI belong to the references above.This part of the paper has been revised.
We gratefully appreciate for your valuable comments.
Reviewer 2 Report
General comments
This interesting work reports on the experimental investigation of the horizontal cycle loading tests on the pile foundation of a 1:50 scale model of an offshore wind turbine (OWT). The influence of horizontal loading on internal force and displacement is investigated.
The topic is of interest in the field of applied ocean engineering because the investigated support structure is widely adopted in relatively low-depth applications of OWTs [1].
With respect to inland wind farms, OWTs can take advantage from more constant and intense wind forcing, thus making the wind generated power more competitive with respect to conventional exhaustible and high environmental impact sources of energy. Anyway, these complex structures are placed in highly demanding environment and require proper design approach to obtain reliable and cost-effective solutions. Two main design issues of OWTs are the uncertainty of environmental forcings (mainly wind and wave) [2] and the non-linear interactions (e.g. wind-wave combination) [3].
The manuscript is clearly written.
The literature review must be improved with the suggested references that highlight related key aspects in the OWT's design and analysis: characterization of environmental forcing and non-linear interactions.
The materials and methods are clearly explained, but some improvements are required.
The conclusions are supported by the results, but the advances and limitations of the present research should be clearly pointed out.
Typos should be carefully checked throughout the manuscript.
Before considering the manuscript for publication, it must be thoroughly improved by following the comments provided below.
Major concerns
- Section 1. In the introduction section (see lines 38 through 41) it should be properly clarified that single pile foundation is a technical solution that is conveniently adopted for relatively low water depth applications [1].
- Section 1. The introduction section properly introduces the subject and clearly describes the relevant technical literature. As the work focuses on the geotechnical aspect pertaining the complex pile-soil interaction, it must be at least mentioned those relevant aspects that play a key role in the pile-soil interaction, i.e. the proper definition of external forcing and the non-linear interactions. Related references [2-3] provided below must be included.
- Section 1. The introduction section clearly illustrates the aim of the work (lines 80-81). However, it should be added a description of the advances introduced by the present research with respect to the published works previously cited.
- Section 2. Pore water effects are neglected by using dry sand as the foundation soil in the tests. Furthermore, aluminium is adopted for the scale model structure instead of steel which is commonly adopted for OWT's support structure. Please, motivate how these choices affect the results and the applicability of the proposed formula in Eq.s (3-4) to the design of full scale support structures for OWTs.
- Section 2. Please, provide short explanation of the scaling low adopted.
- Section 4. Advances and possible limitations related to the present study must be pointed out.
Minor concerns
- Line 19: please, clarify that the investigated type of OWT's foundation is adopted in a peculiar water depth condition, that is low-depth.
- Line 30: a blank space is missed in "discussed.In". This typo occurs very frequently throughout the manuscript (see also lines 31; 55; 69; 74, 225, 255 etc.). I suggest careful rereading of the revised manuscript.
- Line 31: please consider rephrasing as follows "formula. Also, the fatigue".
- Table 1: the lower limit of zero millimetres for the particle size range is inconsistent with the particle size distribution curve in Fig. 1. Please, check.
- Lines 113-114: the sentence "The loading point is at the 1200mm above the mud surface" seems inconsistent with information in Fig. 2 where the applied force is represented at 1.8 m above the bottom surface. Please check.
- Tables 2 and 3: the information of the tower and pile thickness should be added in order that the experiment can be repeated.
- Equation (1): please define in the text the meaning of the symbol f_n.
- Figure 6: according to my understanding, the quantity in the abscissa should be the horizontal displacement (relative to the pile diameter D). Please explain the meaning of "change of horizontal displacement".
- Figures 7 and 8: considering the adopted relationship between the cumulative lateral displacement and the number of cycles in Eq. (3), a logarithmic scale could be conveniently adopted in the related figures, perhaps helping to increase their readability at low number of cycles.
- The symbol N adopted for the number of cycles is the same adopted for the unit of force (see also Figures 6 and 9, y-axis). Please remove any ambiguity.
- Equation (5): please define in the text the meaning of the symbol α.
- Line 254: M_100 does not appear in the Eq. (8). Also, in Eq. (8) M_N is equal to two different expressions at the right-hand side: please clarify.
- Line 255: please consider revising as follows "earlier. Here".
- Line 289: please consider revising as follows "small load ratio (ζ_b≦0.3)"
References
[1] DNV (Det Norske Veritas), Offshore Standard: Design of Offshore Wind Turbine Structures (DNV-OS-J101), 2004.
[2] Saptarshi Sarkar, Lin Chen, Breiffni Fitzgerald, Biswajit Basu. Multi-resolution wavelet pitch controller for spar-type floating offshore wind turbines including wave-current interactions. Journal of Sound and Vibration 470 (2020) 115170
[3] S. Manenti et al. Wind-Wave Hindcasting on Offshore Wind Turbine through Coupled Atmospheric and Spectral Models. Proc. ASCE Earth & Space Int. Conf. March 14-17 Honolulu HI. ISBN 978-0-7844-1096-7
Author Response
Dear Reviewer#2:
Thank you for the reviewer#2's letter and for your comments concerning our manuscript entitled “Experimental study on whole wind power structure with innovative open-ended pile foundation under long-term horizontal loading” (ID: sensors-905595). Those comments are all valuable and very helpful for revising and improving our paper, as well as the important guiding significance to our researches. We have studied comments carefully have made correction which we hope meet with approval. Revised portion are marked in red in the paper. The main corrections in the paper and the responds to your comments are as flowing:
Responds to the your comments about major concerns:
1 Section 1. In the introduction section (see lines 38 through 41) it should be properly clarified that single pile foundation is a technical solution that is conveniently adopted for relatively low water depth applications [1].
Response:This part of the paper has been revised, showing in the Line 36.
2 Section 1. The introduction section properly introduces the subject and clearly describes the relevant technical literature. As the work focuses on the geotechnical aspect pertaining the complex pile-soil interaction, it must be at least mentioned those relevant aspects that play a key role in the pile-soil interaction, i.e. the proper definition of external forcing and the non-linear interactions. Related references [2-3] provided below must be included.
Response:This part of the paper has been revised, showing in the Line 72-75.
3 Section 1. The introduction section clearly illustrates the aim of the work (lines 80-81). However, it should be added a description of the advances introduced by the present research with respect to the published works previously cited.
Response:This part of the paper has been revised, showing in the Line 83-84.
4 Section 2. Pore water effects are neglected by using dry sand as the foundation soil in the tests. Furthermore, aluminium is adopted for the scale model structure instead of steel which is commonly adopted for OWT's support structure. Please, motivate how these choices affect the results and the applicability of the proposed formula in Eq.s (3-4) to the design of full scale support structures for OWTs.
Response:
(1)As for the simplification of model test, there must be differences in physical and mechanical properties between dry sand and saturated sand, but many tests show that dry sand is used instead of saturated sand in the test. the relevant correct test phenomena and conclusions can be approximately obtained.
The relevant references are as follows:
[1]Peng J, Clarke B G, Rouainia M. A device to cyclic later loaded model piles[J].Geotechnical Testing Journal,2006,29(4):341
[2]Chan, S. F., Hanna, T. H.. Repeated loading on single piles in sand [J]. Journal of the Geotechnical Engineering Divisin, ASCE, 1980, 106(2): 171-188.
[3]Lee, C. Y. and Poulos, H. G.. Tests on model instrumented grouted piles in offshore calcareous soil [J]. Journal of Geotechnical Engineering, 1991, 117(11): 1738-1753.
[4]Bhattacharya, S. , & Adhikari, S. . (2011). Experimental validation of soil-structure interaction of offshore wind turbines. Soil Dynamics & Earthquake Engineering, 31(5-6), 805-816.
(2)For the selection of model pile materials, the principle of physical condition similarity should be satisfied: stress similarity ratio Cσ, elastic modulus similarity ratio CE, strain similarity ratio Cε, and Cσ=CECε.The Poisson's ratio of 6063 aluminum alloy pipe is close to that of steel pipe.
Therefore, the physical and mechanical parameters of sand and model pile are close to those of the prototype.
5 Section 2. Please, provide short explanation of the scaling low adopted.
Response:By using a certain similarity theory, the model can be simplified and the physical and mechanical properties of the model test can be obtained.For the test, the the scaling low of the model test should be determined according to the maneuverability of the model test. In this experiment, when the diameter of the model pile is less than 5cm, it will be very difficult to install strain gauges and conductors, so the lower limit of the model test is about 1:100.
6 Section 4. Advances and possible limitations related to the present study must be pointed out.
Response:This part of the paper has been revised, showing in the Line 298-302.
Responds to the your comments about minor concerns:
1 Line 19: please, clarify that the investigated type of OWT's foundation is adopted in a peculiar water depth condition, that is low-depth.
Response:This part of the paper has been revised, showing in the Line 20.
2 Line 30: a blank space is missed in "discussed.In". This typo occurs very frequently throughout the manuscript (see also lines 31; 55; 69; 74, 225, 255 etc.). I suggest careful rereading of the revised manuscript.
Response:This part of the paper has been revised.
3 Line 31: please consider rephrasing as follows "formula. Also, the fatigue".
Response:This part of the paper has been revised, showing in the Line 31,32.
4 Table 1: the lower limit of zero millimetres for the particle size range is inconsistent with the particle size distribution curve in Fig. 1. Please, check.
Response:This part of the paper has been revised, showing in the Line 107.
5 Lines 113-114: the sentence "The loading point is at the 1200mm above the mud surface" seems inconsistent with information in Fig. 2 where the applied force is represented at 1.8 m above the bottom surface. Please check.
Response:This part of the paper has been revised, showing in the Line 117.
6 Tables 2 and 3: the information of the tower and pile thickness should be added in order that the experiment can be repeated.
Response:This part of the paper has been revised, showing in the Line 120, 121.
7 Equation (1): please define in the text the meaning of the symbol f_n.
Response:This part of the paper has been revised, showing in the Line 154-155.
8 Figure 6: according to my understanding, the quantity in the abscissa should be the horizontal displacement (relative to the pile diameter D). Please explain the meaning of "change of horizontal displacement".
Response:I mean (change of horizontal displacement) but mistakenly wrote (change of horizontal displacement). This part of the paper has been revised, showing in the Line 169.
9 Figures 7 and 8: considering the adopted relationship between the cumulative lateral displacement and the number of cycles in Eq. (3), a logarithmic scale could be conveniently adopted in the related figures, perhaps helping to increase their readability at low number of cycles.
Response:I'm sorry. I refuse this suggestion. After trying logarithmic coordinates, the original coordinate system is relatively better.
10 The symbol N adopted for the number of cycles is the same adopted for the unit of force (see also Figures 6 and 9, y-axis). Please remove any ambiguity.
Response:This part of the paper has been revised, showing in the Line 168,232.
11 Equation (5): please define in the text the meaning of the symbol α.
Response:This part of the paper has been revised, showing in the Line 258.
12 Line 254: M_100 does not appear in the Eq. (8). Also, in Eq. (8) M_N is equal to two different expressions at the right-hand side: please clarify.
Response:This part of the paper has been revised, showing in the Eq. (8).
13 Line 255: please consider revising as follows "earlier. Here".
Response:This part of the paper has been revised, showing in the Line 256.
14 Line 289: please consider revising as follows "small load ratio (ζ_b≦0.3)"
Response:This part of the paper has been revised, showing in the Line 283.
We gratefully appreciate for your valuable comments.

Reviewer 3 Report
The authors should clarify the following points before it can be accepted for publication in the journal:
- The soil material properties (such as internal friction angle, undrained shear strength …) are important, so the authors should mention this part in the paper.
- Usually, a small scale test requires the illustration of the similarity. It is often to let the two models (real and testing) have the same natural frequency. The author should discuss this issue.
- As shown in Fig.2, a small load, such as 180 N, will cause a large displacement of 0.1D. The author should discuss where comes from the displacement.
- The author get the conclusion: ‘The displacement value of: “3P” is larger than that of “1P”and natural frequency. This reflects 187 that “3P” has a great influence on the whole structure of wind power pile foundation.’. This conclusion should be right, but the reason should be due to the movement of the three blades. Since there was no blades in the experiment, so the authors should discuss the reason to obtain this conclusion.
- The paper showed the bending moments of the structure, and how to obtain them should be mentioned.
- Cyclic loading will cause fatigue of the structure, but this paper does not mention any of the fatigue problem, such as the fatigue damage, location.
- Only experimental results were shown and discussed, and there was no validation of the experimental result. The authors should discuss this issue. If there are big errors in the experiment, all the investigations may not be right.
Author Response
Dear Reviewer#3:
Thank you for the reviewer#3's letter and for your comments concerning our manuscript entitled “Experimental study on whole wind power structure with innovative open-ended pile foundation under long-term horizontal loading” (ID: sensors-905595). Those comments are all valuable and very helpful for revising and improving our paper, as well as the important guiding significance to our researches. We have studied comments carefully have made correction which we hope meet with approval. Revised portion are marked in red in the paper. The main corrections in the paper and the responds to your comments are as flowing:
Responds to the your comments:
1 The soil material properties (such as internal friction angle, undrained shear strength …) are important, so the authors should mention this part in the paper.
Response:As for the simplification of model test, there must be differences in physical and mechanical properties between dry sand and saturated sand, but many tests show that dry sand is used instead of saturated sand in the test. The relevant correct test phenomena and conclusions can be approximately obtained. The relevant references are as follows:
[1]Peng J, Clarke B G, Rouainia M. A device to cyclic later loaded model piles[J].Geotechnical Testing Journal,2006,29(4):341
[2]Lee, C. Y. and Poulos, H. G.. Tests on model instrumented grouted piles in offshore calcareous soil [J]. Journal of Geotechnical Engineering, 1991, 117(11): 1738-1753.
[3]Bhattacharya, S., & Adhikari, S.. (2011). Experimental validation of soil–structure interaction of offshore wind turbines. Soil Dynamics & Earthquake Engineering, 31(5-6), 805-816.
2 Usually, a small scale test requires the illustration of the similarity. It is often to let the two models (real and testing) have the same natural frequency. The author should discuss this issue.
Response:
In order to accurately simulate the dynamic characteristics of the model structure, the actual field loading frequency and natural frequency of OWE pile should be equivalent to that of the model according to a certain relationship. In the Burbo Bank prototype, the frequency ranges of IP and 3P are 0.083-0.217Hz and 0.25-0.65Hz, respectively. The natural frequency of the on-site OWE pile is about 0.292Hz.
|
(1) |
In this model test, the natural frequency of the model is about 9Hz. The 1P and 3P of the model calculated by formula (1) correspond to 2-5.2Hz (1P) and 6-15.6Hz (3P), respectively. In this paper, the frequency of 1P load is 6Hz and the frequency of 3P load is 12Hz, all of which are in the range of 1P-3P.
Reference:
[1]Bhattacharya S, Lombardi D, Wood DM. Similitude relationships for physical modelling of monopile-supported offshore wind turbines[J]. International Journalof Physical Modelling in Geotechnics. 2011,11(2):58-68.
[2]Chan, S. F., Hanna, T. H.. Repeated loading on single piles in sand [J]. Journal of the Geotechnical Engineering Divisin, ASCE, 1980, 106(2): 171-188.
3 As shown in Fig.2, a small load, such as 180 N, will cause a large displacement of 0.1D. The author should discuss where comes from the displacement.
Response:This part of the paper has been revised, showing in the Line 159-161.
4 The author get the conclusion: ‘The displacement value of: “3P” is larger than that of “1P”and natural frequency. This reflects 187 that “3P” has a great influence on the whole structure of wind power pile foundation.’. This conclusion should be right, but the reason should be due to the movement of the three blades. Since there was no blades in the experiment, so the authors should discuss the reason to obtain this conclusion.
Response:In the loading process of OWE pile, the main source of load is the wind load, while the load caused by the blade is secondary. But the frequency generated by the movement of the OWE pile is huge to the overall structure. Therefore, combined with these two factors, we choose the loading mode selected in this experiment, that is, the frequency and the loading force are coupled together.It can be verified from the following references that the test device simplified to no blades is feasible.
[1]Bhattacharya S, Lombardi D, Wood DM. Similitude relationships for physical modelling of monopile-supported offshore wind turbines[J]. International Journalof Physical Modelling in Geotechnics. 2011,11(2):58-68.
[2]Zaaijer MB. Foundation modelling to dynamic behaviour of offshore wind turbines[J]. Applied Ocean Research. 2006,28(1):45-57.
[3]Wu C, Hong Y, Yan Y, Chang B. Soil-nonwoven geotextile filtration behavior under contact with drainage materials[J].Geotextiles and Geomembranes. 2006,24(1):1-10.
[4] Uscilowska, A. and Kolodziej, J. A.. 1998. Free vibration of immersed column carrying tip mass. Journal of Sound and Vibration 216(1): 147-157. DOI: 10.1006/jsvi.1998.1694.
5 The paper showed the bending moments of the structure, and how to obtain them should be mentioned.
Response:This can be found in Section 3.2 of the article, as shown below:
In the model tests, the strain gauges are installed on both sides (The connection positions of the two strain gauges are perpendicular to the loading direction), and the strain gauges on both sides satisfy the pure bending at the same horizontal height. According to the relevant theories in material mechanics, the bending moment at any point can be determined by
|
(5) |
|
|
(6) |
|
|
(7) |
where I is the moment of inertia of the pile section; r is the distance between the strain measuring point and the neutral layer, that is, the radius; EI is the bending stiffness of the pile. M is the bending moment of the pile;ε is the strain of the pile, and ε1 (the tension side parallel to the loading direction) and ε2(The compressed side parallel to the loading direction) are the strains on both sides of the pile at the same height.
6 Cyclic loading will cause fatigue of the structure, but this paper does not mention any of the fatigue problem, such as the fatigue damage, location.
Only experimental results were shown and discussed, and there was no validation of the experimental result. The authors should discuss this issue. If there are big errors in the experiment, all the investigations may not be right.
Response:Thanks for your comments again.In the experiment, because of the large number of cycles, each time consumes a lot of manpower and material resources. Due to the length of the paper, we have to focus on the some issues, but we are sorry to being incapable of making a good and further analysis of the overall fatigue characteristics of the structure.
We gratefully appreciate for your valuable comments.

Round 2
Reviewer 1 Report
All remarks are adequately addressed no further comments from my side.
Author Response
Dear Reviewer#1:
Thank you for the reviewer#1's letter and for your comments concerning our manuscript entitled “Experimental study on whole wind power structure with innovative open-ended pile foundation under long-term horizontal loading” (ID: sensors-905595).
The paper has been revised again.
We gratefully appreciate for your valuable comments.

Reviewer 2 Report
The Authors have improved the original manuscript following my recommendations.
Before considering it for publication, the Authors are kindly requested to:
- include missing blank spaces throughout the manuscript. See for instance: line 30 ( are discussed.Inconclusion, moment); line 31 (formulas.And,the fatigue); line 36 (in shallow water(DNV,2004) have); line 72 (soil-structure is reduced.The); line 74 (Manenti,2010; 74 Saptarshi,2020).Later,) etc.
- include into the manuscript the explanations provided in the responses to Reviewer major concerns at points 4 and 5 with related references.
Author Response
Dear Reviewer#2:
Thank you for the reviewer#2's letter and for your comments concerning our manuscript entitled “Experimental study on whole wind power structure with innovative open-ended pile foundation under long-term horizontal loading” (ID: sensors-905595).
The paper has been revised again.
We gratefully appreciate for your valuable comments.

Reviewer 3 Report
Please check Figure 6, where the unit of vertical axis and the p value are much different. I do not have further questions.
Author Response
Dear Reviewer#3:
Thank you for the reviewer#3's letter and for your comments concerning our manuscript entitled “Experimental study on whole wind power structure with innovative open-ended pile foundation under long-term horizontal loading” (ID: sensors-905595).
The paper has been revised again.
We gratefully appreciate for your valuable comments.
